# Anatomical Description of Rhinoceros Iguana (*Cyclura cornuta cornuta*) Head by Computed Tomography, Magnetic Resonance Imaging and Gross-Sections

**DOI:** 10.3390/ani13060955

**Published:** 2023-03-07

**Authors:** Eligia González Rodríguez, Mario Encinoso Quintana, Daniel Morales Bordon, José Guerra Garcés, Himar Artiles Nuez, José Raduan Jaber

**Affiliations:** 1Hospital Clínico Veterinario, Facultad de Veterinaria, Universidad de Las Palmas de Gran Canaria, Trasmontaña, Arucas, 35413 Las Palmas, Spain; 2Departamento de Patología Animal, Producción Animal, Bromatología y Tecnología de los Alimentos, Facultad de Veterinaria, Universidad de Las Palmas de Gran Canaria, Trasmontaña, Arucas, 35413 Las Palmas, Spain; 3Rancho Texas Lanzarote Park, Lanzarote, 35510 Islas Canarias, Spain; 4IVC Evidensia Los Tarahales, 35013 Islas Canarias, Spain; 5Departamento de Morfología, Facultad de Veterinaria, Universidad de Las Palmas de Gran Canaria, Trasmontaña, Arucas, 35413 Las Palmas, Spain

**Keywords:** computed tomography, magnetic resonance imaging, gross-sections, reptiles, anatomy, head, rhinoceros iguana

## Abstract

**Simple Summary:**

The rhinoceros iguana (*Cyclura cornuta cornuta*), like many other iguana species, has become severely endangered as a result of human activities, earning the highest level of protection under the CITES convention for protected species. Moreover, it has been classified as vulnerable by the International Union for Conservation of Nature (IUCN). The rhinoceros iguana belongs to the family Iguanidae and is native to the island of Hispaniola. Since its population has declined, the Canary Island Government has promoted an initiative for its recovery in a completely natural environment. The limited literature regarding the anatomy of this species motivated us to investigate its cranial structures by using modern imaging techniques such as computed tomography (CT) and magnetic resonance imaging (MRI) combined with anatomical dissection, to obtain helpful information on the structures that form the rhinoceros iguana’s head.

**Abstract:**

In this paper, we attempted to elaborate on an atlas of the head of the rhinoceros iguana, applying modern imaging techniques such as CT and MRI. Furthermore, by combining the images acquired through these techniques with macroscopic anatomical sections, we obtained an adequate description of the relevant structures that form the head of this species. This anatomical information could provide a valuable diagnostic tool for the clinical evaluation of different pathological processes in iguanas such as abscesses and osteodystrophy secondary to nutrient imbalances, skull malformations, fractures, and neoplasia.

## 1. Introduction

The rhinoceros iguana (*Cyclura cornuta cornuta*), belonging to the Iguanidae family, is an endemic species of the island of Hispaniola, in the Caribbean Sea [1]. Its skin has rough epidermal scales, and its greyish-brown or olive coloring camouflages very well with the environment. The name comes from the bony protuberances or pseudo-horns dorsal to the snout. Males have larger body dimensions than females, and their horns are also larger. This iguana has a polygamous mating system and is oviparous (5–20 eggs per clutch), reaching maturity between 5 and 9 years of age [2]. This species is extremely territorial. To intimidate conspecifics or predators, these animals move their heads and neck, while rotating their bodies. These movements are also used in mating rituals. Like all iguanas, this species is heliothermic, which means it must organize its activities to use solar radiation to regulate its body temperature. This also means that it can only survive in tropical/subtropical climates. Its diet is herbivorous, although it may occasionally feed on insects, land crabs or carrion. Due to human encroachment and destruction of the environment, it has been severely displaced from its habitat (near the coast but with low rainfall). Limestone mining, predation, as well as deforestation by the wood industry and forest fires have forced this species to migrate or even become extinct, as is the case with the closely related subspecies *Cyclura cornuta onchiopsis*, native to Navassa Island [3].

The great interspecific anatomical complexity between mammals and reptiles and the growing interest in reptiles as companion animals present a challenge to veterinary clinicians in diagnostic imaging studies interpretation [4]. Diagnostic imaging has brought a radical change in their clinical practice due to the facility to obtain information on the internal structures of the animal body [5]. Thanks to technological developments in this field, morphological and anatomical information can be obtained in a less invasive and considerably faster way, and this technique has become a fundamental tool for this practice, using conventional radiology [6] and ultrasound [7], as well as more advanced imaging techniques such as CT [8] or MRI [9], which offer certain advantages over conventional ones, such as overlapping structures avoidance, fast image acquisition and high contrast resolution, among others [10,11]. All this has meant that the veterinary clinician has to become acquainted not only with these techniques but also with the anatomy and physiology of reptiles [12].

Some relevant literature on the anatomical, physiological and pathological study of these species is already widely available [8,9,10,11,12,13,14,15]. As far as we know, the anatomy of different reptile species has already been thoroughly described employing diagnostic imaging techniques [13,14,15]. Authors have reported atlases of green, loggerhead and leatherback sea turtles [15,16,17,18], komodo dragons [19,20] and green iguanas [12,15,21,22], snakes and lizards [21]. Concerning the rhinoceros iguana, most of the studies are focused on population genetics [23], evolution and historical biogeography [24], and on some pathologic descriptions such as osteopetrosis-like conditions [25]. However, to the authors’ knowledge, none have performed anatomic investigations on the head of the rhinoceros iguana. Therefore, this study aimed to describe the normal anatomy of the head of this species, using CT, MRI and those anatomic sections that are the more informative regions of gross anatomy and can help to understand all the structures that form the head of the rhinoceros iguana. The application of CT, MRI and macroscopic anatomical sections could provide essential information for anatomic descriptions in teaching and clinical practice.

## 2. Materials and Methods

### 2.1. Animals

Two carcasses of adult female rhinoceros iguanas (*Cyclura cornuta cornuta*) from the zoological park “Rancho Texas Lanzarote Park” (Lanzarote, Canary Islands, Spain) were collected. One female measured 94 cm, and the other was 91 cm long from snout to tail. They weighed 5 and 4.2 kg, respectively. The animals died due to natural causes. No abnormalities were found on physical examination.

### 2.2. Anatomic Evaluation

We performed anatomical gross-sections to facilitate the identification of structures observed in the CT and MRI images. Immediately after the scanning procedures, these specimens were placed in a plastic isolation holder in ventral recumbency and successively stored in a freezer (−80 °C) until completely frozen. Subsequently, the two frozen carcasses were sectioned using an electric band saw to obtain sequential anatomical gross-sections. Contiguous 1 cm transverse slices were obtained starting at the snout and extending to the first cervical vertebra region. These slices were thicker than those for CT and MRI to preserve integrity and position of the anatomic structures in the sections. These sections were cleaned with water, numbered and photographed on the cranial and caudal surfaces.

Afterwards, we selected those anatomic sections that better matched the CT and MRI images to identify the relevant structures of the rhinoceros iguana head. To help in this process, we also used anatomical texts and relevant references describing this species [4,21,26,27].

### 2.3. CT Technique

Transverse CT images were obtained at the Veterinary Hospital of Las Palmas de Gran Canaria University using a 16-slice helical CT scanner (Toshiba Astelion, Canon Medical System, Tokyo, Japan). The animals were placed in ventral recumbency as symmetrically as possible on the CT couch. A standard clinical protocol (100 kVp, 80 mA, 512 × 512 acquisition matrix, 1809 × 858 field of view, a spiral pitch factor of 0.94 and a gantry rotation of 1.5 s) was used to obtain sequential transverse CT images (1 mm thickness). The original transverse data were recorded and transferred to the CT workstation. No CT density or anatomic variations were detected in the head of the reptiles used in the investigation. In this study, we applied two CT windows by adjusting the window widths (WW) and window levels (WL) to appreciate the CT appearance of the head structures: a bone window setting (WW = 1500; WL = 300) and a soft tissue window setting (WW = 350; WL = 40). Moreover, dorsal and sagittal multiplanar reconstructed (MPR) images were also obtained to better visualize other iguana head structures.

### 2.4. MRI Technique

MRI images were obtained with a Canon Vantage Elan 1.5 T equipment, using T1W sequences in a transversal plane (TR: 634 ms, TE: 10, FOV: 1809 × 829, thickness 2 mm, matrix 192 × 160), T2W sequences in a transversal plane (TR: 4769 ms, TE: 120, FOV: 1809 × 829, thickness 2 mm, matrix 192 × 224), T2W sequences in the dorsal plane (TR: 5271 ms, TE: 120, FOV: 1809 × 829, thickness 2.5 mm, matrix 240 × 192), T2W sequences in the sagittal plane (TR: 4450 ms, TE: 120, FOV: 1809 × 829, thickness 2.9 mm, matrix 224 × 224); enhanced spin-echo sequences were performed in the dorsal, transverse and sagittal planes. The MRI images were obtained with a thickness of 2.7–3.5 mm.

## 3. Results

Figure 1 corresponds to a bone-window CT sagittal plane, in which each line and number (I–V) represents approximately the level of the following anatomical, CT and MRI transverse planes. Transversal sections revealing the relevant anatomical structures of the rhinoceros iguana head are presented (Figure 2, Figure 3, Figure 4, Figure 5 and Figure 6). Figure 2, Figure 3, Figure 4, Figure 5 and Figure 6 are composed of three images: (A) macroscopic, (B) bone window CT and (C) MRI. The images are presented in a rostrocaudal progression from the maxillary bone (Figure 2) to the brainstem levels (Figure 6). Figure 7 and Figure 8 are composed of two images: (A) bone-window CT and (B) MRI in T2W, in a dorsal and sagittal plane, respectively. Figure 9 is a T2W sequence sagittal image showing the angulation concerning the horizontal axis of the myelencephalon (40°).

### 3.1. Anatomical Sections

Different structures belonging to the central nervous system were visualized by anatomical gross-sections. Therefore, we identified the brain (telencephalon) and the two telencephalic hemispheres (Figure 4A and Figure 5A) separated by the *fissura longitudinalis cerebri* (Figure 5A), the diencephalon (thalamus) (Figure 4A and Figure 5A), the dorsal part of the mesencephalon with the two caudal colliculus (Figure 6A) and the ventral part of the cerebellum with the nodule, covering part of the fourth ventricle, as well as the ventral surface of the brainstem (myelencephalon) (Figure 6A). These sections were also helpful for the observation of the olfactory bulb that showed an extracranial location, ventral to the frontal bones, and coursed between the two eyeballs (Figure 3). In addition, these sections allowed the description of structures of the eyeball, identifying the cornea, the sclera, the retina, the vitreous humor, the lens, as well as associated structures such as the interorbital septum and the orbital sinus (Figure 3A and Figure 4A). In addition, these transverse sections allowed the identification of structures belonging to the oral cavity such as the tongue (Figure 2A) and other structures such as the larynx, with the corniculate tubercles of the arytenoid cartilage, the thyroid cartilage, and the laryngeal ventricle (Figure 3A), as well as of different structures of the nasal cavity such as the nasal glands that filled almost the entire cavity, the conchal grooves and the stammteil located laterally to the nasal septum (Figure 2A). Adjacent structures such as the trachea and the nasopharynx were also well identified (Figure 4A, Figure 5A and Figure 6A). Furthermore, most bony structures that form the neurocranium were observed, such as the pterygoid, frontal, postfrontal-postorbital, parietal, supraoccipital, basioccipital, exoccipital, otoccipital, parabasisphenoid and sphenoid bones (Figure 3A, Figure 4A, Figure 5A and Figure 6A), as well as those that form the splanchnocranium such as the nasal, vomer, palatine and maxillary bones (Figure 2A and Figure 3A) and also the medial horn located dorsal to the nasal bone (Figure 2A), the mandible, with the dentary bone (Figure 3A), and the hyoid apparatus, visualizing the central body of the hyoid arch, between the lateral branches of the dentary bone (Figure 3A). Rostromedially to the two dentary bones, we identified different muscles related to the hyoid apparatus such as the *musculus genihyioideus, hyoglossus and intermandibularis* (Figure 2A and Figure 3A). In the following sections, we also observed the muscle groups corresponding to the mandibular musculature (we were not able to dissect them, and therefore they were treated as a group), including the *pterygoideus, omohyoideus, sternohyoideus, ceratohyoideus*, *adductor mandibulae externus medialis* and its homonym *superficialis* (Figure 4A and Figure 5A).

### 3.2. Computed Tomography (CT)

Regarding the neurocranium, the CT images allowed us to distinguish bone structures such as the prefrontal, frontal, postfrontal-postorbital, parietal, squamosal, quadrate, epipterygoid, pterygoid, basioccipital, exoccipital, otoccipital and parabasisphenoid bones (Figure 3B, Figure 4B, Figure 5B and Figure 6B); related to the splanchnocranium, we observed the nasal, premaxilla, maxilla, septomaxilla, vomer, jugal and palatine bones, and the nasal septum (Figure 2B, Figure 3B, Figure 4B, Figure 7B and Figure 8B); the mandible structures such as the dentary, angular, surangular, coronoid and articular bones (Figure 2B, Figure 3B, Figure 4B, Figure 5B and Figure 6B) and the hyoid apparatus (Figure 2B, Figure 3B, Figure 4B, Figure 5B and Figure 6B). CT scanning and post-processing transverse images showed the relation between the different bones that form the head of the rhinoceros iguana, the junction of the nasal and the prefrontal bone, as well as that of the parietal bone with the postfrontal-postorbital bone (Figure 2B and Figure 4B), and the palatine and quadrate processes of pterygoid bone (Figure 5B and Figure 6B respectively). The prominent medial horn was identified dorsal to the nasal bone with soft-tissue attenuation and a thin, lamellar-shaped mineral structure bordering it regularly on its most external aspect (Figure 2B).

Concerning the nasal cavity, the transverse CT image showed the nasal glands as symmetrical bilateral structures, with regular and well-defined margins, located on both sides of the nasal cavity and with soft tissue attenuation (Figure 2B). Moreover, those structures with intraluminal gas content such as the nasal conchal recess (Figure 2B), oral cavity, nasopharyngeal duct, trachea, adductor fossa (Figure 4B) and the otic cavity (Figure 6B) were identified with this technique, appearing with a vacuum effect. In addition, there were areas of soft tissue attenuation medial to the mandible and bilateral to the hyoid apparatus, compatible with the *intermandibularis*, *genihyioideus* and *hyoglossus* muscles (1 in Figure 2B and Figure 3B), the *pterygoideus*, *omohyoideus*, *sternohyoideus* and *ceratohyoideus* muscles (2 in Figure 4B), the *adductor mandibulae externus medialis* and *superficialis* muscles located, respectively, dorsomedially and ventrolaterally to the adductor fossa (3 and 4 in Figure 4B and Figure 5B).

In addition, we distinguished with adequate resolution different structures of the central nervous system such as the olfactory bulb, the brain (telencephalon and diencephalon), the cerebellum and the brainstem (Figure 3B, Figure 4B, Figure 5B and Figure 6B).

### 3.3. Magnetic Resonance Imaging (MRI)

The soft structures of the iguana’s head, such as the central nervous system as well as the eyeball’s structures (vitreous humor and lens), the oral cavity with the tongue and the masticatory muscles, showed an accurate visualization using MRI (Figure 2C, Figure 3C, Figure 4C, Figure 5C and Figure 6C). Therefore, an increased volume of both eyeballs in proportion to the size of the head was seen in all sequences (Figure 3C and Figure 7B). As in CT, the structures with gas content (Figure 2C and Figure 6C) appeared with a vacuum effect, being hypointense in all sequences. The nasal glands were bilaterally symmetric, with regular and well-defined margins, located on both sides of the nasal cavity, being iso/hyperintense in T1W and T2W sequences, compared to the encephalic grey matter (Figure 2C and Figure 4C). In contrast, the medial horn appeared hypo/isointense on T1W and T2W sequences concerning the white matter and with mild differentiation of the external bony cortex in relation to the white matter (Figure 2C).

In contrast to the CT images, the bone junctions were not distinguishable on MRI, but those bones that formed the neurocranium, such as the frontal, postfrontal-postorbitary, parietal and supraoccipital bones were identified (Figure 3C, Figure 4C, Figure 5C, Figure 6C and Figure 8B). The cranial musculature was found isointense concerning the thalamus in T2W. This technique enabled a better resolution to identify the muscle groups already mentioned (Figure 2C, Figure 3C, Figure 4C and Figure 5C). In the transverse planes of the encephalon, the cerebral cortex was observed slightly more hyperintense than the white matter, which was more hypointense in T2W sequences (Figure 4C). The diencephalic region (Figure 4C, Figure 5C and Figure 8B) was hypointense (T2W) compared to the cerebral cortex (Figure 5C), showing the thalamus and hypothalamus (Figure 8B). The brainstem appeared hypo/isointense in T2W compared to the cerebral cortex, as well as presenting a markedly tortuous horizontal alignment (Figure 8B). Moreover, the caudal colliculus and the fourth ventricle were also displayed in excellent detail. In the rostral aspect of the telencephalon, we distinguished the dorsal pallium rostral part with its lateral and medial portions (Figure 4C). Interestingly, the dorsal MRI image was quite helpful to identify the olfactory bulb located extracranially, which extended rostromedially between the eyeballs (Figure 3C, Figure 7B and Figure 8B).

## 4. Discussion

Technological developments in imaging techniques have improved the anatomical knowledge and the diagnosis of several pathologies. From conventional imaging methods, such as radiography and ultrasound, to advanced ones such as CT and MRI, the level of resolution, the rapid acquisition of images and the absence of superimposition have meant innovation in research, daily clinical practice and academic purposes [11,12,28,29,30].

According to other descriptive studies of different reptile species [1,3,12,14,16,17,20], the images obtained by CT, MRI and gross-sections were adequate to study the rhinoceros iguana head. Therefore, the gross-sections provided accurate anatomic characteristics of the head structures, mainly those related to the brain, the eyeball and the larynx. To the authors’ knowledge, a unique description using gross-section and conventional CT equipment was performed of the green iguana head [22], but only a few brain and laryngeal structures were labelled.

Considering the green iguana [26], some differences were found between the bony structures of these species. Thus, we identified the presence of three horns on the dorsal aspect of the nasal region, which are absent in the green iguana. Moreover, the evaluation of the transverse CT images obtained in the bone window and the post-processing in MPR showed a lower prominence of the occipital ridge in the rhinoceros iguana compared to the green iguana. In addition, as described by other authors [22], the CT images of the rhinoceros iguana head displayed in excellent detail the bony structures compared to the anatomic gross-section. With MRI, the palatine, frontal, postfrontal-postorbital, parietal, supraoccipital bones and crest were distinguishable due to a gross-section and with both imaging techniques. In addition, the eyes were easily displayed with all the techniques used. Nonetheless, specific eyeball structures, such as the iris and the scleral ossifications, were better identified by CT and were hardly visible in anatomic gross-sections, where the lens, the retina, the sclera and the vitreous chamber could be visualized. Some of these structures are of scientific and morphological interest to perform further studies on the dimensions of the eyeball and vision ability [29,30].

The use of MRI was valuable for visualizing organs located in the head. Thus, transverse MR images T2W facilitated the accurate identification of the main components of the brain, such as the telencephalon, diencephalon, mesencephalon, metencephalon and myelencephalon. Therefore, we identified specific structures such as the lateral ventricles, the anterior dorsal and posterior dorsal ventricular ridges, the cerebellum, the fourth ventricle, the dorsal pallium and the brain stem with the caudal colliculus. Interestingly, the dorsal MR T2W image showed the olfactory bulb isointense compared to the telencephalon, located in the rostral portion of the encephalon but in an extracranial situation, remaining in the rostromedial aspect of the eyeballs (extracranial structures). In contrast to other articles that excluded the olfactory bulb description, we present the shape, location and intensity of this structure [31]. This extracranial location was already observed in previous studies on the tawny dragon [31]. However, this finding had not been reported in other species of reptiles, such as the rhinoceros iguana. The gross-section images confirmed the presence of these structures and their rostral extension. Interestingly, in other species, such as galliform birds, the olfactory bulbs are separated from the telencephalic hemispheres but located intracranially, protected by the frontal bone [32].

In addition, the MRI and CT findings showed a greater angulation concerning the horizontal axis of the encephalon, with an angulation of 40° between the central horizontal axis of the skull and the central axis of the encephalon (dorsal displacement concerning the encephalon of the dog) (Figure 9), differing from other studies where an angulation of 28° was described [31]. In galliform birds, the brain has a similar angulation to the skull axis [32]. More similar aspects to the latter were found, such as the large size of the eyeballs, which were almost as large as the whole encephalon [32]. Nonetheless, further studies with a large number of animals should be performed to confirm these findings.

Finally, several studies carried out in reptiles have been performed with micro-CT equipment, as it has a higher resolution, allowing a better distinction of anatomical formations [33,34,35]. However, in our work, the combined application of different advanced techniques (CT, MRI) and macroscopic anatomical sections provided helpful information for anatomic descriptions in teaching and clinical practice.

## 5. Conclusions

In conclusion, CT and MRI appeared to be adequate tools, providing anatomical references of the different bone and soft tissue structures comprising the head of the rhinoceros iguana. Therefore, the findings obtained in this study are helpful for evaluating numerous processes involving the head of these animals, such as abscesses, metabolic bone diseases, fractures and neoplasia. Moreover, these two imaging techniques could contribute to veterinary anatomy learning by our students as these techniques allow the visualization of structures without overlapping, eliminating the difficulties of visualizing the extent of different types of lesions.

## Figures and Tables

**Figure 1 animals-13-00955-f001:**
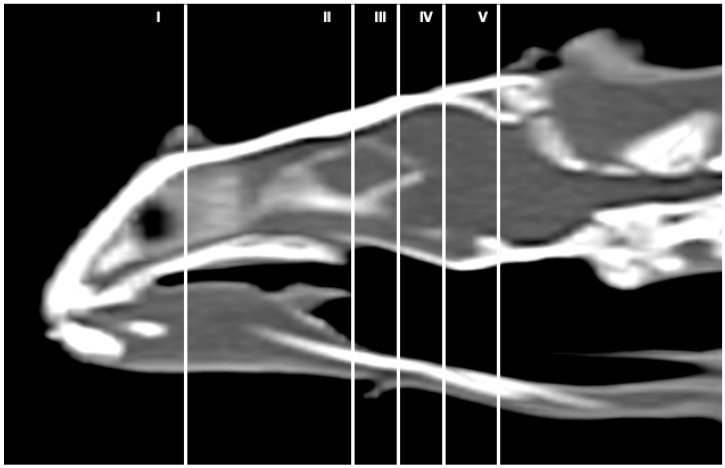
Sagittal CT bone window image of the head of rhinoceros iguana that approximately representing the level of the slices of this study. Segments I–V correspond to Figure 2, Figure 3, Figure 4, Figure 5 and Figure 6.

**Figure 2 animals-13-00955-f002:**
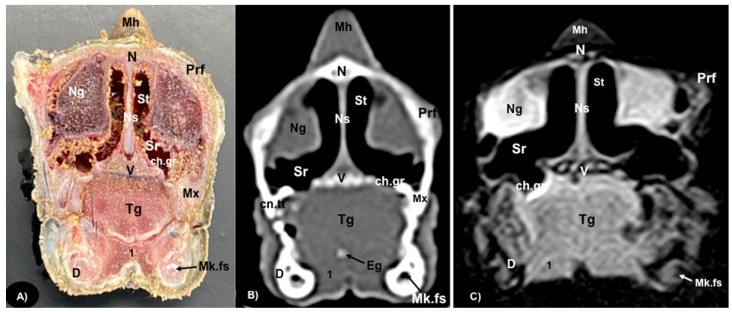
Transversal gross-section (**A**), CT bone window (**B**), and MRI in T1W (**C**), images of the head of rhinoceros iguana at the level of the nasal cavity corresponding to line I in Figure 1. Mh: Medial horn. N: Nasal bone. Prf: Prefrontal bone. Ns: Nasal septum. Ng: Nasal glands. Sr: Subconchal recess. St: Stammteil. ch.gr: Choanal groove. V: Vomer. Mx: Maxillary bone. Tg: Tongue. Eg: Entoglossal process of the basihyal bone. cn.tt: Canine tooth. 1: *Musculus geniohyoideus + Musculus hyoglossus + intermandibularis*. D: Dentary bone. Mk.fs: Meckelian fossa.

**Figure 3 animals-13-00955-f003:**
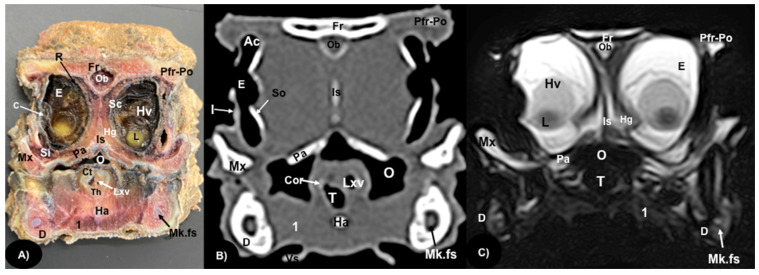
Transversal gross-section (**A**), CT bone window (**B**), and MRI in T2W (**C**), images of the head of rhinoceros iguana in the orbital region corresponding to line II in Figure 1. Fr: Frontal bone. Pfr-Po: Postfrontal-postorbital bone. Ob: Olfactory bulb. Ac: Anterior chamber. R: Retina. So: Scleral ossicles. I: Iris. E: Eyeball. C: Cornea. Sc: Sclera. L: Lens. Hv: Vitreous chamber. Is: Interorbital septum. Hg: Harderian gland. Si: Infraorbital sinus. Pa: Palatine bone. Mx: Maxillary bone. O: Oral cavity. Cor: Corniculated tubercles of arytenoid cartilage. Lxv: Laryngeal ventricle. Th: Thyroid cartilage. D: Dentary bone. Mk.fs: Meckelian fossa. Ha: Hyoid apparatus. T: Trachea. 1: *Musculus geniohyoideus + Musculus hyoglossus + Musculus intermandibularis*. Vs: Ventral spines.

**Figure 4 animals-13-00955-f004:**
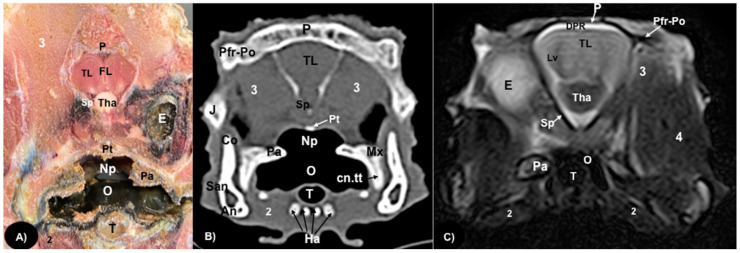
Transversal gross-section (**A**), CT bone window (**B**), and MRI in T2W (**C**), images of the head of rhinoceros iguana at the level of the frontal bone corresponding to line III in Figure 1. 3: *Musculus adductor mandibulae externus medialis*. P: Parietal bone. Pfr-Po: Postfrontal-Postorbital bone. TL: Telencephalon. FL: *Fissura longitudinalis cerebri*. DPR: Dorsal Pallium Rostral Part. Tha: Thalamus. E: Eyeball. Sp: Sphenoid bone. Pt: Pterygoid bone. J: Jugal bone. Pa: Palatine bone. O: Oral cavity. 4: *Musculus adductor mandibulae externus Pars superficialis.* Mx: Maxillary bone. cn.tt: Caniniform tooth. Np: Nasopharyngeal duct. Co: Coronoid. San: Surangular bone. An: Angular bone. 2: *Musculus intermandibularis + Musculus geniohyoideus + Musculus hyoglossus + Musculus pterygoideus + Musculus omohyoideus + Musculus sterohyoideus + Musculus ceratohyoideus.* T: Trachea. Ha: Hyoid apparatus.

**Figure 5 animals-13-00955-f005:**
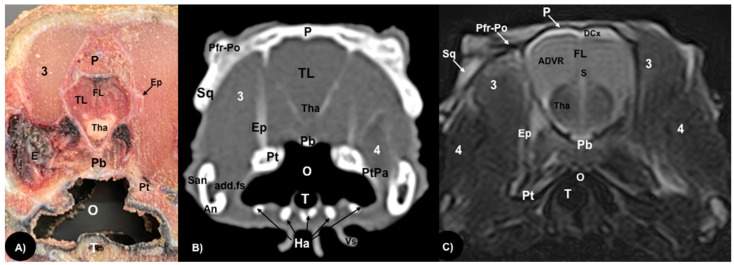
Transversal gross-section (**A**), CT bone window (**B**) and MRI in T2W (**C**), images of the head of rhinoceros iguana at the level of the anterior dorsal ventricular ridge (ADVR) corresponding to line IV in Figure 1. P: Parietal bone. Pfr-Po: Postfrontal-Postorbital bone. Sq: Squamousal bone. E: Eyeball. TL: Telencephalon. FL: *Fissura longitudinalis cerebri.* Tha: Thalamus. ADVR: Anterior Dorsal Ventricular Ridge. DCx: Dorsal cortex. 3: *Musculus adductor mandibulae externus medialis*. 4: *Musculus adductor mandibulae externus superficialis*. Pb: Parabasisphenoid bone. Pt: Pterygoid bone. Ep: Epipterygoid bone. PtPa: Pterygoid-palatine processes. add.fs: Adductor fossa. An: Angular bone. San: Surangular bone. Ha: Hyoid apparatus. O: Oral cavity. T: Trachea. Vs: Ventral spines.

**Figure 6 animals-13-00955-f006:**
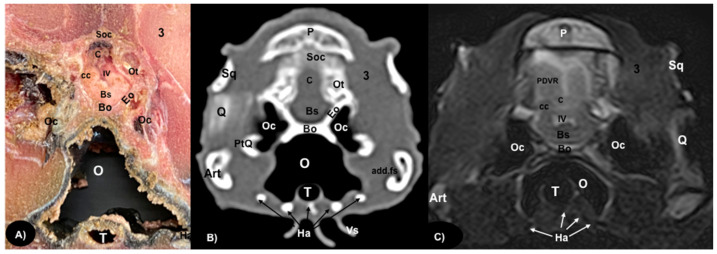
Transversal gross-section (**A**), CT bone window (**B**) and MRI in T2W (**C**), images of the head of rhinoceros iguana at the level of the occipital region corresponding to line V in Figure 1. P: Parietal bone. Soc: Supraoccipital bone. Sq: Squamosal bone. 3: *Musculus adductor mandibulae externus superficialis*. C: Cerebellum. PDVR: Posterior Dorsal Ventricular Ridge. Bs: Brain stem. cc: Caudal colliculus. IV: Fourth ventricle. Ot: Otoccipital bone. Eo: Exoccipital bone. Bo: Basioccipital bone. Oc: Otic cavity. Q: Quadrate bone. PtQ: Pterygoid-quadrate processes. Art: Articular bone. add.fs: Adductor fossa. O: Oral cavity. T: Trachea. Ha: Hyoid apparatus. Vs: Ventral spines.

**Figure 7 animals-13-00955-f007:**
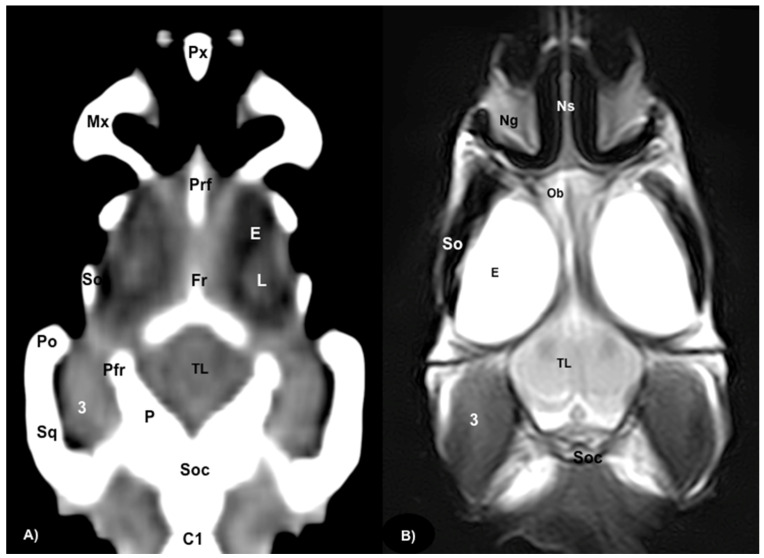
(**A**) Dorsal CT image in the brain window. (**B**) Dorsal MRI in the T2W sequence of the head of rhinoceros iguana telencephalon and at the olfactory bulb level. Px: Premaxillary bone. Ns: Nasal septum. Mx: Maxillary bone. Ng: Nasal glands. Ob: Olfactory bulbs. Prf: Prefrontal bone. E: Eyeball. L: Lens. So: Scleral ossicles. Fr: Frontal bone. TL: Telencephalon. 3: *Musculus adductor mandibulae externus medialis*. Pfr: Postfrontal bone. Po: Postorbital bone. P: Parietal bone. Soc: Supraoccipital bone. Sq: Squamosal bone. C1: First cervical vertebra.

**Figure 8 animals-13-00955-f008:**
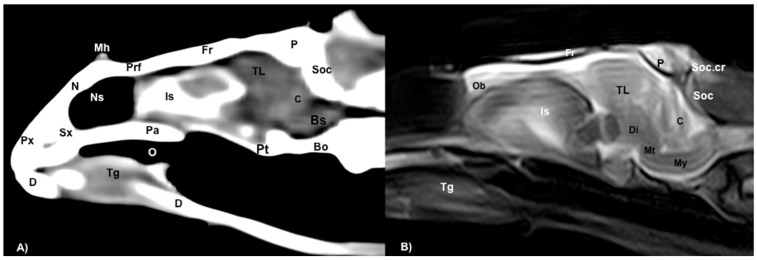
(**A**) Mid-sagittal CT image in the brain window. (**B**) Sagittal MRI in the T2W sequence image of the head of rhinoceros iguana. Px: Premaxillary bone. Sx: Septomaxilla. D: Dentary bone. N: Nasal bone. Mh: Medial horn. Prf: Prefrontal bone. Pa: Palatine bone. O: Oral cavity. Tg: Tongue. Fr: Frontal bone. Is: Interorbital sinus. P: Parietal bone. Soc: Supraoccipital bone. Soc.cr: Supraoccipital crest. Pt: Pterygoid bone. Bo: Basioccipital bone. Ob: Olfactory bulbs. TL: Telencephalon. C: Cerebellum. Bs: Brainstem. Di: Diencephalon. Mt: Metencephalon. My: Myelencephalon.

**Figure 9 animals-13-00955-f009:**
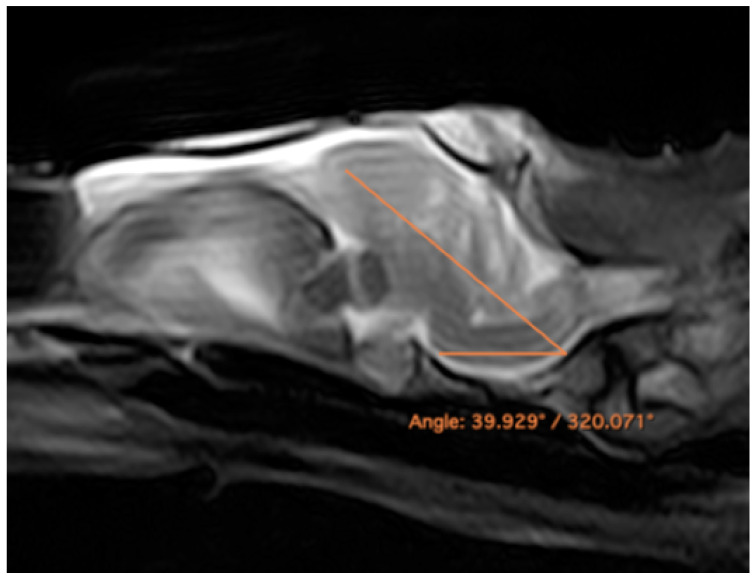
Rhinoceros iguana encephalic angulation, sagittal T2W image with respect to the horizontal axis of the myelencephalon (40°).

## Data Availability

Data is unavailable due to privacy or ethical restrictions.

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
