# Peer review of "Anatomical Description of Rhinoceros Iguana (Cyclura cornuta cornuta) Head by Computed Tomography, Magnetic Resonance Imaging and Gross-Sections"

_animals, 2023, doi:10.3390/ani13060955_

Round 1

Reviewer 1 Report

·       The manuscript used CT, MRI, and cadaver cross-sections to describe the anatomy of the rhinoceros iguana head.

·       The author must do her best to increase the detailed description of many structures; please cite and enhance the paper in comparison with Banzato, T., Selleri, P., Veladiano, I. A., Martin, A., Zanetti, E., & Zotti, A. (2012). Comparative evaluation of the cadaveric, radiographic and computed tomographic anatomy of the heads of green iguana (Iguana iguana), common tegu (Tupinambis merianae) and bearded dragon (Pogona vitticeps). BMC Veterinary Research, 8(1), 1-11.

·       live adult animals must be used to CT and mri not dead animals

·       The medial horn in Figure 2 did not describe the results.

·       Please cite all structures mentioned in the figures and legends in the results.

·       The resolutions in the CT and MRI images are not good; try to capture new, high-resolution photos.

·       Some references missed such as Banzato, T., Selleri, P., Veladiano, I. A., Martin, A., Zanetti, E., & Zotti, A. (2012).

Author Response

Reviewer 1 comments for Author:

  1. The author must do her best to increase the detailed description of many structures; please cite and enhance the paper in comparison with Banzato, T., Selleri, P., Veladiano, I. A., Martin, A., Zanetti, E., & Zotti, A. (2012). Comparative evaluation of the cadaveric, radiographic and computed tomographic anatomy of the heads of green iguana (Iguana iguana), common tegu (Tupinambis merianae) and bearded dragon (Pogona vitticeps). BMC Veterinary Research, 8(1), 1-11.

Following your recommendation, the suggested paper has been considered to increase the detailed description of the iguana head structures. Moreover, we have included and discussed that information in the discussion section

  1. Live adult animals must be used to CT and MRI not dead animals

We completely agree with this comment, but considering that these are endangered animals and the risk involved in submitting them to general anaesthesia is always important, we have carried out these scans on animals that have unfortunately died due to natural causes and always with the consent of the person responsible of the reserve.

  1. The medial horn in Figure 2 did not describe the results.

Thank you very much for your comment. As you suggested, we added the description of the medial horn from the CT and MRI images.

  1. Please cite all structures mentioned in the figures and legends in the results.

Following your proposal, we have included all the structures of the figures in the results.

  1. The resolutions in the CT and MRI images are not good; try to capture new, high-resolution photos.

As you recommend, we have captured new, high-resolution CT and MRI images, which show better anatomic detail.

  1. Some references were missed such as Banzato, T., Selleri, P., Veladiano, I. A., Martin, A., Zanetti, E., & Zotti, A. (2012).

Thank you for your contribution. This reference was used for the identification of several structures of the rhinoceros iguana head, and it is included in the revised version of the manuscript.

Reviewer 2 Report

I`m sending my suggestion for Authors:

1. Simple summary - the sentence "...normal anatomy of this specie has encouraged..." - please make a correction of misspelling the word "specie"

2. Abstract - see above comment about the misspelling

3. Introduction - see above comment about the misspelling

4. Material and methods section - "MRI technique" - "2,5 mm; 2,9 mm; 2,7-3,5 mm" - the commas should be replaced with dots

5. Figure 8 - within the legend of this figure please make a correction: "Hipothalamus" as: "hypothalamus"

6. The conclusion should be added at the end of the discussion.

Author Response

Reviewer 2 comments for Author:

  1. Simple summary - the sentence "...normal anatomy of this specie has encouraged..." - please make a correction of misspelling the word "specie"

As you suggested, we have corrected the misspelling along with the simple summary.

  1. Abstract - see the above comment about the misspelling

As you suggested, we have corrected this misspelling. Moreover, we have revised and refined the abstract.

  1. Introduction - see the above comment about the misspelling

As you suggested, we have corrected the misspelling in the introduction.

  1. Material and methods section - "MRI technique" - "2,5 mm; 2,9 mm; 2,7-3,5 mm" - the commas should be replaced with dots

This suggestion has been corrected.

  1. Figure 8 - within the legend of this figure please make a correction: "Hipothalamus" as: "hypothalamus"

This suggestion has been corrected. Therefore, we have replaced "Hipothalamus" by "hypothalamus"

  1. The conclusion should be added at the end of the discussion.

As you recommend, we have added the conclusion section at the end of the discussion.

Reviewer 3 Report

This paper presents a much-needed anatomic description of dated literature on the head anatomy of iguanids. Although the authors focused their attention on the value of this contribution to veterinary clinicians, it does have a lot of value for those who study the evolution of cranial structures in reptiles and, more broadly, in vertebrates.

My comments pertain to the way the paper is presented and the way it is structured, as I think it is partly downplaying the value of their findings by focusing too much on the technical aspects of imaging and their value. It is by now very clear that CT and MRI are valuable tools to study anatomy, and not only micro-CT should be used to image structure. Furthermore, it emphasises the value of having dissections presented alongside CT/MRI images.

I have attached a pdf with more comments, but broadly these are my general criticisms:

Introduction

The introduction is too short and focuses mostly on the advantages/benefits of having CT/MRI images for veterinary clinicians. I was expecting, however, to have a review of what literature has been produced on rhinoceros iguanas, what pieces of work are dated and maybe a bit on the reasons for choosing the sections figured. Based on the abstract, I am under the impression that these sections are the ones where certain pathologies are more commonly diagnosed. However, it can also be because those are the more informative regions of gross anatomy. Either way, I expected any of this to be explained in the introduction. One of the last paragraphs of the conclusions actually reads more appropriately as an introduction.

Results

I had several issues with the figures in the text, mostly regarding the labelling and the captioning. Comments are included in the attachment. A general comment on this, maybe the authors should use more standard abbreviations for structures, such as the ones used in Evans (2008) [Ref. 14 in the authors's paper].

Discussion

The Discussion section of this paper needs to be refocused. Currently, there is a great deal of why CT and MRI are good options for studying anatomy, but there is not much mention of how the results illustrate this. Some of this is mentioned in the Results section but should be brought up and expanded here.

Secondly, this section will greatly improve if there is a comparison between the gross anatomy and the imaging techniques. What elements of the gross anatomy are not visible in the CT/MRI and vice versa?

It can also be good to compare the anatomy of the rhinoceros iguana with the green iguana and the tawny dragons from some of the references cited.

Finally, I think that the Discussion should be focused more on the anatomical part that currently is only seen in the figures and less on how Imaging Techniques can be useful. After Lauridsen et al. (2011)'s paper, it is clear that CT/MRI are good alternatives to studying anatomy, and there are several examples of this in the Reference list. There is no need to make this case anymore.

What is new is that the authors provide an atlas of images of rhinoceros iguanas that add to a mostly dated literature on the matter. The Introduction and the Discussion should reflect this novelty. I am looking forward to seeing this published.

Author Response

Reviewer 3 comments for Author:

Firstly, we really appreciate the comments provided by the reviewer since we believe they have served to improve the quality of our manuscript.

  1. The introduction is too short and focuses mostly on the advantages/benefits of having CT/MRI images for veterinary clinicians. I was expecting, however, to have a review of what literature has been produced on rhinoceros iguanas, what pieces of work are dated and maybe a bit on the reasons for choosing the sections figured. Based on the abstract, I am under the impression that these sections are the ones where certain pathologies are more commonly diagnosed. However, it can also be because those are the more informative regions of gross anatomy. Either way, I expected any of this to be explained in the introduction. One of the last paragraphs of the conclusions actually reads more appropriately as an introduction.

First, we would like to thank you for all the feedback you gave us. We consider it very positive for the improvement and soundness of our work. As you suggest, we have expanded and modified the structure of the introduction, first focusing on the biology of this species and its endangered condition and then including literature on the use of diagnostic imaging techniques to study the anatomy of different reptile species. We highlight that only sparse reports have been published concerning biology, pathology and the anatomy of the rhinoceros iguana, which was our main purpose.

  1. Results

I had several issues with the figures in the text, mostly regarding the labelling and the captioning. Comments are included in the attachment. A general comment on this, maybe the authors should use more standard abbreviations for structures, such as the ones used in Evans (2008) [Ref. 14 in the authors's paper].

Thank you very much for your comment. There were many errors, and we believe we have corrected them by using the standard abbreviation described by Evans (2008), as well as using other reports (The head and Neck Anatomy of Sea Turtle and Skull Shape in Testudines. Jones et al., 2012).

  1. Discussion

The Discussion section of this paper needs to be refocused. Currently, there is a great deal of why CT and MRI are good options for studying anatomy, but there is not much mention of how the results illustrate this. Some of this is mentioned in the Results section but should be brought up and expanded here.

As you suggested, we have expanded this section, explaining the visualization and resolution of different structures by CT and MRI, and how the images obtained in our work were adequate for studying anatomy. Moreover, we compared the visualization of the main anatomic formations with these techniques

Secondly, this section will greatly improve if there is a comparison between the gross anatomy and the imaging techniques. What elements of the gross anatomy are not visible in the CT/MRI and vice versa?

Following your recommendation, we have redone the discussion section, including a comparison between the gross anatomy and the imaging techniques, specifying those elements that are clearly visible with each technique

It can also be good to compare the anatomy of the rhinoceros iguana with the green iguana and the tawny dragons from some of the references cited.

As you recommend, in this revised version of the manuscript, we have compared some specific detail with those species and focused more on the anatomical detail

Round 2

Reviewer 1 Report

Thank you for your response. The paper now became better

Author Response

Dear Editor,  

Thank you very much for your very positive feedback on our work. Indeed, we have tried to correct these final mistakes as you recommended. 

Sincerely

Raduan Jaber